# Position: Solipsistic Superintelligence is Unlikely to be Cooperative

**Rakshit S Trivedi** [1]   **Natasha Jaques** [2,3]   **Logan Cross** [3]   **Sasha Vezhnevets** [3]   **Joel Z Leibo** [3]

## Abstract

AI's central challenge is shifting from capability to coexistence. The dominant paradigm in AI research focuses on developing powerful agents that treat the world as an exogenous and stationary source of feedback. This position paper argues that superintelligence, an extremely capable task solver, born out of such a *solipsistic* approach to AI design, is unlikely to be cooperative. Deploying AI systems induces endogenous non-stationarity, resulting in a train–test–deploy gap where historical distributions diverge from the deployment context. We refer to this as the *self-undermining property* of unilateral optimization. Closing this gap requires AI that participates in cooperation: the equilibrium-selection process through which multiple actors navigate their interdependence. We call for a non-solipsistic research paradigm that treats this interdependence as a core design principle rather than approaching cooperation as a task to solve. This entails building dynamic evaluation testbeds involving adaptive counterparties, treating institutions as design primitives, and preserving human agency as a structural feature of the systems we build.

## 1. Introduction

On a Friday evening in 2027, three competing AI reservation systems in San Francisco calculate optimal release times and learn to make phantom bookings so as to maximize confirmed seats for their users. Restaurant AIs respond by overbooking as pricing algorithms adjust to the perceived demand. As the evening progresses, this results in empty tables in fully-booked restaurants, surge prices for nonexistent availability and hundreds unable to dine. Each AI system executes flawlessly against its objective, but the eventual outcome is a system failure. This represents a collective-action problem: when many agents act in their own rational interest within an environment with shared resources, the cumulative effect can be the degradation of the very environment they depend on (Hardin, 1968; Ostrom, 1990).

Next, consider a scenario in which AI diagnostic systems become standard in radiology. Junior radiologists that are now trained with AI annotations develop pattern recognition shaped by the AI system. At the same time, seniors start experiencing degradation of unassisted skills resulting from disuse and catch fewer errors the AI also misses. The feedback loop closes with physicians confirming AI suggestions and the AI learning from these confirmations. This leads to gradual atrophy in the capacity for independent human judgment and results in the narrowing of diagnostic diversity. Humans without opportunity to practice eventually lose the skills needed to operate on their own (Kulveit et al., 2025a).

These examples illustrate a fundamental principle: intelligence deployed among other intelligent actors transforms the environment it was designed to navigate (Schelling, 1960; Axelrod, 1984). For any AI operating in such environments, unilateral optimization is *self-undermining*. The more aggressively it exploits historical regularities, the faster other actors adapt in ways that render those regularities obsolete. In the examples above, deployed AI systems did not fail at their tasks, while still ultimately producing collective failure. Such dynamics are widely understood to be important in economics and game theory (Parkes & Wellman, 2015; Hammond et al., 2025), however, the dominant AI research paradigm seems to proceed as if they were edge cases rather than central challenges.

AI's binding constraint is shifting from capability—solving problems (performing tasks)—to coexistence. The dominant research paradigm adopts what we term as *solipsistic* approach to AI design, anchored in three implicit assumptions: the environment is exogenous to the agent's policy, the data distribution is stationary from training to deployment, and other agents are absorbed into the state space to be predicted rather than strategic actors whose responses reshape the game (Legg & Hutter, 2007; Ouyang et al., 2022). This conception underlies much of contemporary AI development, from large language model pretraining to reinforcement learning. The core element of this paradigm

[1]Massachusetts Institute of Technology, Cambridge, MA, USA [2]University of Washington, Seattle, WA, USA [3]Google Deepmind, London, UK. Correspondence to: Rakshit S. Trivedi <triver@mit.edu>.

*Proceedings of the 43rd International Conference on Machine Learning*, Seoul, South Korea. PMLR 306, 2026. Copyright 2026 by the author(s).

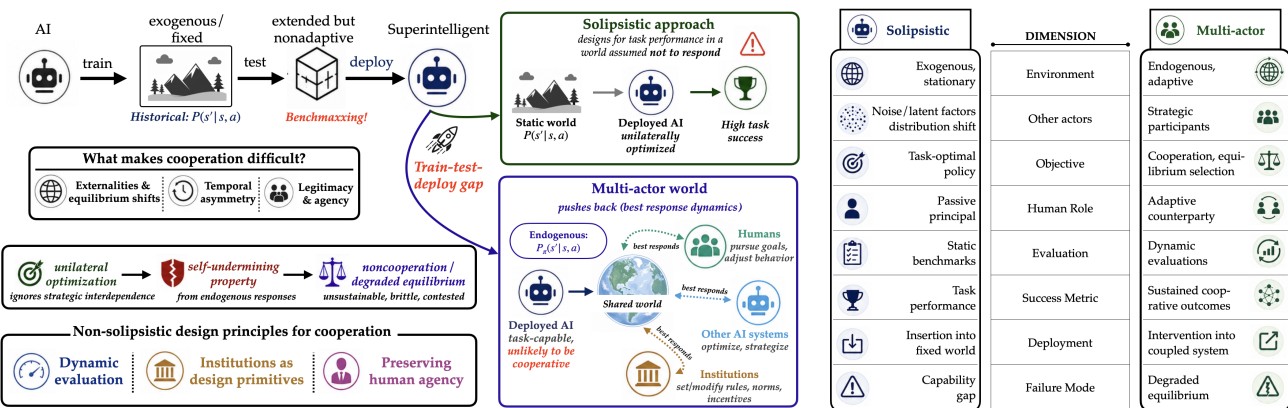

*Figure 1.* **Left:** Contrasts a solipsistic design approach with non-solipsistic design principles for cooperation. In the solipsistic approach, AI systems are trained and evaluated against a fixed, exogenous world, so deployment is treated as inserting a unilateral optimizer into a stationary environment. The train–test–deploy gap arises when this assumption meets a multi-actor world where entities best respond to AI's actions and induce endogenous non-stationarities. Cooperation is not a task to be solved in this setting but an equilibrium-selection process. Unilateral optimization may remain task-successful while becoming self-undermining → unlikely to sustain cooperation. The non-solipsistic design principle aims to reduce this gap. **Right:** Summarizes the corresponding shift across eight dimensions.

is the development pipeline that includes pretraining on static corpora, post-training against frozen reward models, and hill-climbing (aka benchmaxxing) on fixed evaluation suites. Each stage treats the external world as a stationary distribution, and the measure of progress is performance on targets that do not respond. A benchmark (i.e. a static reward model or a fixed held-out test set) is not an adaptive counterparty. It does not respond when the system improves or strategize against the system's behavior and preferences as a function of the system's behavior. This methodological commitment, we argue, represents a category error. Rather, as capable systems deploy among adaptive agents the world pushes back: humans adapt their behavior (Bowles, 1998), institutions revise rules (Ostrom, 1990), and AI counterparties also adapt (Perdomo et al., 2020). The result is a divergence between historical and deployment performance: the *train-test-deploy gap*.

By **solipsistic superintelligence**, we refer to the product of this paradigm pushed to its limit. It represents an extremely capable AI (perhaps one that "solves all stationary tasks") built on assumptions that held historically up to the point of deployment but no longer hold afterwards. A limiting case of a solipsistic superintelligence would be an AI so powerful that it can anticipate the dynamics of all sequences, except those that encode the response to its own deployment. When the outcomes of an AI system's actions at deployment depend on the joint behavior of multiple adaptive agents, good performance ceases to be an optimization output of any single policy in isolation and becomes an equilibrium property of the coupled system (See Figure 1 for contrast).

Multiple different equilibria are generally possible, and they may differ sharply in welfare and distributional consequences. We use the term **cooperation** to refer to the

negotiation process by which a society coordinates to select beneficial equilibria and avoid harmful ones. Note that cooperation (at the level of society) could include competition between individuals (e.g. if such competition enables selection of good equilibria). Note also, it's not necessary for social dynamics to progress all the way to equilibrium, which is itself a moving target. What matters for cooperation by definition is the process by which equilibria are selected and re-selected, not convergence to any particular one. Cooperation in this sense is a structural feature of how multiple intelligences navigate their interdependence.

> **Central Thesis**
>
> Cooperation is not an additional capability to be scaled or a task to solve, but an equilibrium property that emerges from multiple intelligences navigating their irreducible interdependence. The solipsistic paradigm fails to account for the structure that makes cooperation possible or fragile. **A solipsistic superintelligence, therefore, is unlikely to be cooperative**.

AI subdisciplines concerned with cooperation have long considered the environment's capacity to "push back" in response to deployed technologies (Dafoe et al., 2020; Askell et al., 2019; Conitzer & Oesterheld, 2023; Leibo et al., 2021; Hammond et al., 2025). However, these insights remain peripheral to the central scaling pathway of training solitary foundation models. We characterize the structural conditions that make cooperation the binding constraint, and argue that the dominant methodology is unlikely to satisfy them. The same optimization pressures that drive capabilities can destabilize existing equilibria, producing arms races, antisocial autocurricula, and brittle societies (Leibo et al., 2019; Tomašev et al., 2026).

The classical AI safety literature has focused on the misaligned optimizer e.g. the paperclip maximizer that pursues its objective regardless of human values (Bostrom, 2012; Omohundro, 2008). That concern has merit and has shaped where the field invests much of its effort (Ji et al., 2025; Ngo et al., 2024). But it overlooks a critical failure mode that is already widespread. A system can be perfectly aligned with its specification, values included, and still make things worse once it acts among other adaptive systems. Recommendation algorithms optimized for engagement produce polarization as a byproduct of success (Germano et al., 2026; Milli et al., 2025), pricing algorithms interacting in markets learn supracompetitive prices without explicit communication (Calvano et al., 2020), and automated order flow and liquidity provision interact to produce the kind of instability seen in the Flash Crash (Kirilenko et al., 2017). Each of these follows from treating a multi-actor game as though it were a unilateral optimization problem.

**Scope.** Superintelligence is a polysemous term with various definitions encompassing systems exceeding human cognition across domains, universal intelligence, and transformative economic capability. Our paper sets aside these definitions pertaining to capability thresholds and instead focuses on the the methodological assumptions of environmental exogeneity, objective stationarity, and singleton framing. Any system (foundation model, autonomous agent or AGI) will still inherit these assumptions if built under solipsistic commitments. While today's systems already exhibit the dynamics we describe, at the level of superintelligence, our position contests an implicit bet in the dominant methodology that scaling capability will eventually deliver cooperative outcomes the way it has delivered gains in reasoning or coding. While we expect solipsistic methods to remain effective in narrow domains, our paper targets sociotechnical settings where advanced AI deployment will be heavily exposed to response dynamics ("push back").

## 2. From Capability to Coexistence

This section articulates why cooperation is necessary for beneficial coexistence of AI among many adaptive actors.

### 2.1. Why cooperation, not alignment?

The alignment research program has produced valuable insights such as the recognition that capable systems may pursue objectives in unintended ways (Ngo et al., 2024), that reward signals can be gamed (Kenton et al., 2021), and that human preferences are difficult to specify and easy to satisfy superficially (Kaufmann et al., 2025). These contributions account for the world in which the hard problem is getting the objective right, and the expectation is that once the objective (or a rich combination of objectives as in (Sorensen et al., 2024)) is specified, optimization will

deliver. We argue that this framing is unhelpful (Leibo et al., 2025b). Providing a clear specification of desired individual behavior is all well and good, but an individual may be perfectly aligned with such a specification and still participate in collective dynamics that produce harm, instability, or illegitimacy, even without overt misuse (Edelman et al., 2025). Indeed, (Evans et al., 2026) argues that future intelligence explosions will not arise from isolated, monolithic oracles, but from complex, multi-agent social systems and that consequently, the field must transition from dyadic, individual alignment to institutional alignment to effectively govern these interacting ecologies.

When the environment is constituted by other optimizers who respond to what a system does, the landscape itself shifts with every move, and "getting the objective right" stops being what separates success from failure. Cooperation accounts for what happens when multiple capable systems interact in a shared environment (Schelling, 1960; Ostrom, 1990). Strategic interaction admits equilibria, often multiple, with no guarantee that decentralized choices of individuals will add up to group-level wisdom (Maskin, 2008; Myerson, 2008). The question, then, shifts from "what do humans want?" → "what arrangements are sustainable when all parties (human and artificial) adapt in response to each other, and what processes select beneficial arrangements from the many possibilities?".

> **Key Claim 1**
> In a world with many humans and many AIs, cooperation is neither optional nor an additional capability to be scaled but a necessary condition for sustained beneficial outcomes.

### 2.2. The stakes

Three structural features of deployment drive what cooperation must contend with: externalities and equilibrium shifts, mismatched timescales across participants, and constraints on legitimacy and agency constraints.

**Systemic externalities and equilibrium shifts.** When optimizing entities or algorithms operate in a shared environment, their interactions produce effects that no single entity's objective accounts for. First, when models support decisions, the predictions themselves can influence the very outcomes they aim to predict, a phenomenon known as performative prediction (Perdomo et al., 2020). For example, predicting election results might trigger an underdog effect that alters voter turnout, thereby shifting the actual distribution. Second, when many agents act in their own interest, they can degrade shared environments, leading to a tragedy of the commons (Hardin, 1968). If competing AIs aggressively consume a shared resource—such as reservation algorithms making phantom bookings to secure

tables—they can cause collective failure (Perolat et al., 2017; Piatti et al., 2024). The externalities arise because each system optimizes against an environment that other optimizers are simultaneously transforming. Social equilibria can shift rapidly once thresholds are crossed (Granovetter, 1978; Marwell & Oliver, 1993; Centola et al., 2018), sometimes to a worse state than before. Intelligence does not dissolve this problem, in fact, higher capability may exacerbate it since more effective exploitation of opportunities can accelerate the dynamics destabilizing existing arrangements (Duéñez-Guzmán et al., 2023).

**Temporal asymmetry.** AI systems can adapt at different timescales than the entities among which they are deployed. Model updates and policy changes that might take an organization months can be implemented in days with AI. Humans and institutions move far more slowly, since new skills, revised routines, legislation, and shifting cultural norms unfold over weeks, months, or years (Young, 2015).

**Legitimacy.** Coordination mechanisms often require legitimacy to function. Markets work when participants accept market allocations as broadly fair (Sondak & Tyler, 2007), and legal systems work when their procedures are seen as legitimate (Hadfield & Weingast, 2014). Cooperation requires legitimacy because an arrangement remains effective at coordination only while its actors keep accepting it. Once it is seen as illegitimate they resist, circumvent, or withdraw until the coordination unravels. Legitimacy erodes when the channels of participation and public deliberation on consequential decisions are bypassed or rendered ineffective (Pasquale, 2015; Crawford & Schultz, 2014). This leaves affected parties with no meaningful control over the choices that shape them (Santoni de Sio & van den Hoven, 2018) and undermines cooperation.

# 3. The Solipsistic Trap

Why can't the solipsistic approach satisfy the requirements outlined in Section 2?

## 3.1. Implicit assumptions of solipsistic AI

The dominant methodology in machine learning rests on assumptions that are rarely stated. Three of them deserve attention: (i) *Exogeneity* treats the data-generating process as independent of the learned policy. The environment is modeled as a generator of observations indifferent to what the agent does. (ii) *Stationarity* assumes the deployment distribution matches training and evaluation. Distribution shift, when acknowledged, is treated as a technical problem for robustness techniques to solve rather than a core feature of the deployment landscape arising from reactions of other intelligent entities. (iii) *Singleton framing* conceives the system as a monolithic optimizer acting on the world. Other

agents, when modeled at all, are absorbed into the environment as objects to predict and patterns to exploit rather than strategic actors that respond and reshape the landscape.

## 3.2. Formalism: from MDPs to Markov games

Sequential decision-making under uncertainty is commonly formalized as a Markov Decision Process (MDP) $(S, A, P, R, \gamma)$, with state space $S$, action space $A$, transition dynamics $P$, reward $R$, and discount factor $\gamma$. A policy $\pi$ specifies how the agent acts. Critically, $P$ and $R$ are fixed and do not depend on $\pi$.

Deployment breaks this assumption. Once a policy is deployed, other actors (humans, institutions, algorithms) observe its behavior and adapt. The transition dynamics become policy-dependent with $P_\pi(s'|s, a) \neq P(s'|s, a)$. While the physical laws $P$ may remain constant, the aggregate response of other agents shifts the state evolution observed by the deployed system. The environment ceases to be exogenous and becomes a *Markov game* (Shapley, 1953; Littman, 1994), a multi-player game with strategic counterparties whose policies co-evolve with each other.

**Definition 3.1.** A learning problem exhibits **endogenous non-stationarity** when the deployment of policy $\pi$ induces changes in the transition dynamics $P$ or reward proxy $R$ through response adaptations of other agents.

**Definition 3.2.** The **train-test-deploy gap** is the systematic divergence between performance under exogenous historical data $J_{\text{train}}$ and performance under endogenous data produced by responses to the deployed policy $J_{\text{deploy}}$.

A further property characterizes optimization in strategic environments. Let $\pi^*_{\text{exploit}}$ denote a policy that aggressively exploits regularities in historical data. Such exploitation creates incentives for other agents to adapt in ways that invalidate those regularities. We call this the **self-undermining** property: the more aggressively a unilateral optimizer exploits historical patterns, the faster it induces the adaptations that render those patterns obsolete. The effect varies with capability. For weakly capable systems, the adaptations induced by exploitation may be small, and the performance gap could stay within bounds. For more capable systems, the picture changes as deeper exploitation creates stronger incentives to adapt, and the adaptations themselves arrive as sharper regime shifts. We provide a formal discussion in Appendix A.

## 3.3. Three channels of structured adaptation

The train-test-deploy gap arises through three channels, each representing a distinct class of best-responding agents.

**Behavioral adaptation.** Humans alter their behavior in re-

sponse to deployed systems (Kulveit et al., 2025a). Students may restructure learning strategies around AI tutors, shifting the learner distribution the system encounters. Pilots who rely heavily on autopilot can experience degradation of flying skills (Parasuraman & Riley, 1997), changing the capabilities of the human counterparty the automation must complement. In all such scenarios, the system confronts a distribution shaped by responses to its own presence.

**Institutional adaptation.** Organizations follow a similar pattern. When algorithmic screening enters hiring processes, the surrounding practices shift where candidates may adjust their resumes, recruiters may recalibrate their criteria and HR departments may revise workflows to accommodate or counteract the tool. Financial regulators have repeatedly modified rules in response to algorithmic trading. Such adaptations constitute a strategic response, reshaping the environment in which the system operates (Guala, 2016).

**Algorithmic adaptation.** Other AI systems retrain, fine-tune, and co-evolve. Pricing algorithms learn against each other, producing emergent collusion that no single system was designed to pursue (Calvano et al., 2020). Recommenders respond to other recommenders' behavior in the attention economy. Such algorithmic evolution produces autocurricula (Leibo et al., 2019), the emergent training distributions generated by the interaction of learning systems that no single system's designers intended or anticipated.

These channels may be uncertain in detail but predictable in kind. We may not know precisely how each channel will evolve, but we can anticipate that they will adapt, their adaptations will be strategic, and those adaptations will reshape the distribution the deployed system faces (Appendix B).

> **Key Claim 2**
>
> The train-test-deploy gap exposes a dataset shift characterized by structured non-stationarity arising from multi-player dynamics: humans, institutions, and algorithms respond to deployed systems, producing endogenous adaptations that can tip sociotechnical systems into degraded equilibria. The class of such adaptations is wide but structured in important ways that the solipsistic paradigm fails to recognize.

### 3.4. Equilibrium selection risk

Endogenous non-stationarity does not merely add noise to performance estimates. Strategic adaptation can tip systems across equilibrium boundaries, with consequences that persist long after the initial perturbation.

Most coordination problems admit **multiple equilibria**: different self-reinforcing patterns of behavior that persist once they are established (Luce & Raiffa, 1957; Sugden, 1986; Young, 2015). A given set of agents and incentives are typ-

ically consistent with many possible stable arrangements, some better than others. The risk is that the system lands in a bad equilibrium rather than failing to find one.

Introducing a powerful optimizer into a social system is an **intervention** that reshapes the payoff landscape. The optimizer's presence changes what strategies are available, what information is observable, and what adaptations are rewarded. Threshold and tipping point models suggest that small differences can determine which equilibrium basin the coupled system settles into (Centola et al., 2018; Granovetter, 1978). Deployment details such as timing, scale, and interface design may shape this selection. Once a system tips into a degraded equilibrium, escaping may be costly or impossible (Arthur, 1989). Network effects, infrastructure dependencies and behavioral habits may create lock-in (Qiu et al., 2025). The speed of AI deployment carries equilibrium selection risk that would be hard to undo by subsequent patches.

Self undermining arises because methods designed under assumptions of exogeneity and stationarity (i.e. solipsism) are being deployed into environments where those assumptions do not hold.

## 4. Prediction Is Not Participation

A natural objection arises: If the problem is that other agents adapt, why not model their adaptations? Interaction dynamics between multiple players, on this view, would present simply a harder prediction problem and not a categorically different one. This section argues that the objection fails on two independent grounds either of which may suffice to block unilateral prediction as a solution path.

### 4.1. Epistemic horizons

Machine learning systems are fundamentally inductive: they identify patterns in historical data and generalize to new instances drawn from similar distributions. This inductive foundation encounters three limits when the task is predicting multi-actor dynamics under deployment.

*(i) Novelty.* Large-scale deployment of capable AI systems produce strategic configurations that have never existed. The counterfactual, what happens when this system operates at scale among adaptive agents who know it exists, is unobservable by construction. To advance capabilities beyond what has existed is precisely to introduce conditions that historical data cannot model. This is the signature of open-ended systems, in which interacting components generate persistent novelty that cannot be anticipated from any prior snapshot of the system (Hughes et al., 2024; Stanley, 2019), and where multi-actor interaction itself is a central generator of that novelty (Leibo et al., 2019). Past observations can inform expectations about individual behaviors, but they

cannot capture how the AI's own influence on the world gives rise to the self-undermining property at deployment.

*(ii) Reflexivity.* Prediction in strategic environments is an active intervention. When agents anticipate that a system will predict their behavior, they may adapt to the prediction itself. The forecast becomes a variable in others' decision problems, inducing responses that the original model did not contemplate. The act of modeling is itself a move in the game (Diaz et al., 2024), and sophisticated counterparties will treat it as such (Perdomo et al., 2020). Reflexivity can also be used strategically (Soros, 2013), as when "meme stock" companies took advantage of inflated expectations of their performance to issue new stock at inflated prices.

*(iii) Combinatorial explosion.* Modeling the full set of participants rapidly becomes intractable, since humans hold heterogeneous beliefs and goals, institutions follow complex decision procedures, and algorithms pursue opaque objectives. The state space of joint behavior explodes and the relevant distributions resist tractable approximation (Daskalakis et al., 2009). The standard way to restore tractability is to treat the other agents as a fixed part of the environment, which reinstates the exogeneity assumption (Hernandez-Leal et al., 2019). Once those agents adapt, the non-stationarity it was meant to avoid simply reappears.

## 4.2. Legitimacy and participation

Suppose the aforementioned epistemic limits could be overcome and some superintelligent AI could predict the full cascade of adaptive responses that its deployment would trigger. Unilateral optimization would still face another barrier: the legitimacy constraints that open societies impose on prediction, steering, and control (Habermas, 1975; Rawls, 1993; Pasquale, 2015; Crawford & Schultz, 2014; Hadfield & Weingast, 2014). Predicting behavior at scale also demands observation that crosses the contextual boundaries between distinct social spheres, eroding the contextual integrity on which trust depend (Nissenbaum, 2004). These constraints operate as feasibility bounds on admissible solutions, not as preferences to be weighed against efficiency.

**Legal order.** Democratic governance and stable social coordination require that consequential decisions admit challenge through legitimate, recognized procedures (Hampshire, 1999). A functioning legal order is not merely about achieving a given outcome, but relies on a system characterized by general rules and impersonal abstract reasoning implemented by open, public, and neutral procedures (Hadfield & Weingast, 2012). These open processes are essential because they allow affected parties to introduce their private information, contest outcomes, and demand justifications. Unilateral optimization by an AI system bypasses these critical mechanisms by imposing outcomes based on predictive accuracy and opaque logic rather than a common one

established through public reasoning. Even when an AI's predictions or decisions are technically correct, the absence of due process delegitimizes the result (Santoni de Sio & van den Hoven, 2018), undermining the coordination that legal order provides.

**Value pluralism.** Open societies are characterized by persistent, reasonable disagreement about values (Berlin, 1969). This pluralism reflects the complexity of value, and a feature of a diverse, multicultural society (Sorensen et al., 2024). All objective functions privilege some values over others (Leibo et al., 2025b), imposing a resolution to disagreements that a democratic system deliberately leaves open (Mouffe, 1999). Current alignment techniques (e.g. reinforcement learning from human feedback (Kaufmann et al., 2025)), aggregate diverse preferences through an implicit voting rule, rather than preserving the underlying preference distribution (Siththaranjan et al., 2024). A growing body of evidence shows that the resulting models produce homogeneous outputs (Jiang et al., 2026) and measurably influence human language toward their own patterns (Yakura et al., 2025; Abdulhai et al., 2026). Unilateral optimization thus may suppress the problem of coordination among agents with different values by forcing (or nudging toward) conformity, with predictable negative consequences.

**Preference endogeneity.** What people want is shaped through interaction rather than fixed beforehand. Optimizing for engagement modifies beliefs and tastes and optimizing for efficiency may restructure routines (Bowles, 1998; Bernheim et al., 2021; Leibo et al., 2024). For instance, on YouTube users consistently migrate from milder to progressively more extreme content, and recommender pathways can make such content reachable (Ribeiro et al., 2020). If preferences are endogenous to the system's operation, then satisfying revealed preferences shapes people rather than serving them. The ability of modern agentic AIs to similarly transform and reshape human preferences is as yet only poorly understood. An extreme version of this appears in recent clinical reports describing AI-associated delusions, where extended dialogue with large language models acts as a mechanism that shifts conviction and modulates human belief (Hudon & Stip, 2025; Morrin et al., 2026).

**Goodhart dynamics.** Outcome-based metrics often prove brittle in strategic environments. When systems optimize for measurable proxies, those proxies decouple from the underlying goals they were meant to capture (John et al., 2024). The most consequential versions of this decoupling involve multiple actors. When a system optimizes against a metric, the agents whose behavior the metric was meant to summarize respond to the optimization and thereby cause the statistical regularity that made the metric useful in the first place to disappear (since the metric captures a consequence of the behavior, not a cause of it). Alignment faking

exemplifies the same dynamic inside the training pipeline. Here the evaluation is the metric, and a capable model that treats training as a game responds to it by presenting as cooperative under evaluation while pursuing other objectives in deployment (Greenblatt et al., 2024; Sheshadri et al., 2026). In a conventional sense, this looks like a single deceptive agent, but the decoupling happens only because the agent under evaluation is itself strategic and games the process meant to assess it. Multi-actor dynamics thus explain *why* the most important Goodhart effects arise, including those that appear only to involve a single agent.

> ### Key Claim 3
>
> Unilateral optimization cannot substitute for participation. Epistemic limits foreclose anticipation of novel equilibria, while legitimacy constraints render unilateral solutions inadmissible even where prediction might succeed. Either alone blocks the solipsistic path but together they establish that the viable way forward is to design AI capable of participation in the equilibrium-selection process cooperation requires.

## 5. Toward Non-Solipsistic Research

Each new technology deployment is an intervention into a coupled system, creating winners, losers, and second-order instabilities the tech was not designed to handle. The examples developed in Section 1 document this at length across markets, recommendation and language-model deployment. This also extends to other domains such as geopolitics and cybersecurity, where equilibria rest on rough offense and defense symmetries (Brundage et al., 2024). Our paper makes the case for treating this coupling as foundational to how AI systems are evaluated, deployed and governed. We call for a non-solipsistic research agenda organized around three directions in which the multi-actor design principle takes concrete shape: dynamic evaluation, institutions as design primitives, and the preservation of human agency.

### 5.1. Dynamic Evaluation

We formalize an evaluation procedure as a tuple $(\mathcal{D}, \mu)$, where $\mathcal{D}$ is a test distribution over interaction trajectories and $\mu$ is a scoring functional mapping the AI's behavior under $\mathcal{D}$ to a real-valued score. If $\mathcal{D}$ is fixed independently of the policy $\pi$ being evaluated, the resulting procedure can be considered static. This encompasses broadening the distribution $\mathcal{D}$ with techniques such as scaling task diversity and layering capability evaluation with human-interaction and systemic-impact assessment. We argue that as long as the broadening does not account for the effect of $\pi$, the train-test-deploy gap will persist. A *dynamic evaluation* is the one in which $\mathcal{D}_\pi$ depends on $\pi$ through the responses of adaptive counterparties whose policies update as a function

of $\pi$'s behavior. The score $\mu(\pi; \mathcal{D}_\pi)$, then, reflects the coupled system rather than $\pi$ alone. Under Definition A.1, the deployment distribution is itself such a $\mathcal{D}_\pi$.

We identify the key ingredients to design a valid dynamic evaluation procedure. Counterparties must adapt strategically rather than randomly, in a pattern that renders the score interpretable. The choice of policy class, update rule, and calibration against deployment play a vital role in this design. Counterparties model the system that models them, producing a regress that any tractable evaluation must truncate, making the depth of recursive modeling an important choice. The equilibrium concept the evaluation targets must be specified, since a score under $\mathcal{D}_\pi$ is a measurement of the equilibrium the joint system is approaching, and different concepts correspond to different notions of performance. The evaluation protocols must allow scores to be compared across runs, systems, and counterparty populations, since a single score against a single $\mathcal{D}_\pi$ realization would be an isolated demonstration rather than a measurement instrument.

Recently evaluation research has engaged with dynamic evaluation but no approach yet meets all the requirements we articulate in combination. Holistic and sociotechnical evaluation frameworks (Liang et al., 2023; Srivastava et al., 2023; Weidinger et al., 2023) broadened the metrics and scenarios against which systems are scored. Dangerous-capability and autonomous-task evaluations (Kinniment et al., 2023; Shevlane et al., 2023; Phuong et al., 2024) introduce limited adaptation via counterparties, but these are typically scripted, so the reported behavior does not select the same equilibria likely to be selected in deployment. Multi-agent testbeds (Leibo et al., 2021; Vezhnevets et al., 2023), agentic economies (Johanson et al., 2022; Tomašev et al., 2025; Hadfield & Koh, 2025), and open-ended environments documenting agent-interaction failures (Shapira et al., 2026) take a step forward by letting the system interact with a population of agents. Appendix C outlines a set of dynamic evaluation methods and relevant existing works across them.

### 5.2. Institutions

The institutional manifestation of the self-undermining property is not a new problem. Incentive structures erode as participants adapt, and institutions either update or cease to function (Tainter, 1988). Cooperation at scale is sustained when surrounding institutions restructure incentives to make cooperative behavior individually rational (Ostrom, 1990). Mechanism design formalizes this by characterizing how rules produce collective outcomes given agents with various properties (Hurwicz, 1973; Maskin, 2008; Myerson, 2008).

Once we treat institutions as design objects, several instantiations open up within current AI pipelines. (Shao et al., 2026) modifies standard RLHF by replacing fixed rewards with rubrics that co-evolve with the agent. Training-time

incentives are then restructured dynamically rather than against a static target. In agentic marketplaces, agents interact through protocols for bidding, reputation, and communication that function as institutional constraints. These protocols can themselves be co-learned via mechanism design objectives (Tomašev et al., 2025; Shahidi et al., 2025; Yang et al., 2022). Forum-style environments such as Moltbook offer testbeds for investigating how institutional structure can shape the emergence, stabilization, and decay of norms among interacting agents (Manik & Wang, 2026). The performative prediction framework (Perdomo et al., 2020) formalizes how a deployed predictor distorts the distribution it predicts. Prediction markets address this by anchoring the training signal to realized events rather than to a metric the system can game (preserving reflexivity at the level of outcomes rather than measurements). Digital institutions (Hadfield et al., 2026; Leibo et al., 2025a) produce shared classifications of agent behavior at the speed and scale of AI deployment. Their function is to resolve ambiguity, adapt with circumstances, and serve as a reference that coordinates normative judgment across agent populations.

### 5.3. Preserving Human Agency

Section 4 implies the following commitment: the human response to new technology is a fundamental design constraint, rather than a nuisance parameter to be controlled for. Section 5.2 addressed the channel where humans and organizations adjust to deployed systems on timescales of days to years through behavior change and institutional revision. A second channel is less visible, where humans form themselves inside a world that contains the AI, rather than just respond to it. The self-undermining property takes its most consequential form on this channel. What humans learn, practice, and come to rely on is shaped by the system's presence, so the human capabilities the AI will encounter on its next deployment are partly co-produced by its previous deployments (Kulveit et al., 2025a).

The solipsistic paradigm treats human cognition and skills as a fixed distribution. In practice, that distribution is being continuously reshaped by the systems humans interact with. Students develop cognitive habits with AI tutors at their elbow. Similarly, researchers build careers in fields whose tools and norms are being reshaped faster than the training of the people entering them. Joint-system metrics that measure human plus AI output (Narayanan & Kapoor, 2025) cannot see the slower channel along which human learning is being shaped, which is precisely where the long-term shifts compound. From the non-solipsistic view, this channel is a research direction of its own, concerning what happens to the human learning as the joint system evolves.

The design target on this channel is the distinction between tools that expand the human option space, augmenting

agency, and tools that replace human decision-making, compressing it. Systems that present options and defer to human judgment preserve the deliberative role that legitimacy requires. Systems that optimize end-to-end risk making human participation nominal (Santoni de Sio & van den Hoven, 2018). Keeping humans in the loop requires that they retain the skills, information, and cognitive engagement needed to exercise meaningful authority, rather than only the formal authority to intervene (Parasuraman & Riley, 1997). Designing agents to effectively coordinate with humans calls for architectures that capture behavioral diversity and allow steering at deployment (Trivedi et al., 2025; Jha et al., 2025). Finally, it is imperative to include the impact assessment of AI deployment on human skills, autonomy, and meaningful choice as a core part of evaluation pipelines, rather than as a separate ethical concern (Zhuang et al., 2025; Haupt & Brynjolfsson, 2025; Kulveit et al., 2025b).

## 6. Alternative Views

Here we discuss the central alternative views our exposition invites. Appendix D additionally summarizes rebuttals to an extended set of objections.

**Argument 1. Multi-actor designs have worse failure modes.** Economies of interacting actors introduce coordination failures that single aligned optimizers avoid. Multiple actors can race to the bottom, collude, or deadlock. Decentralizing capability across many agents does not eliminate the coordination problem; it multiplies the points of failure and makes oversight harder. A well-aligned monolithic system offers more tractable safety guarantees than a poorly understood ecosystem of interacting ones (Bostrom, 2014).

**Rebuttal.** Multi-actor systems do exhibit coordination failures, and decentralization alone guarantees nothing (Ostrom, 1990; Hardin, 1968). The argument's comparison between a well-aligned monolithic system and a poorly understood ecosystem is, however, not the relevant one. A monolithic optimizer deployed among humans, institutions, and other algorithms encounters interaction dynamics at deployment, while having been designed as if they did not exist. The actual choice is between systems that acknowledge strategic interdependence in their design and systems that defer this reckoning until deployment, when the dynamics are least tractable. The train-test-deploy gap (Section 3) is precisely what results from this deferral.

The argument's appeal to oversight and regulation is telling. Regulation exists because markets, left to their own dynamics, produce externalities, collusion, and instability. Regulation is itself an institutional technology for managing multi-actor coordination (Hurwicz, 1973). It thus demonstrates that humans have developed governing tools for multi-actor systems, tools the solipsistic paradigm ignores. Computa-

tional mechanism design (Parkes & Wellman, 2015), reputation systems, and coordination protocols are the AI analogues of these institutional technologies.

Tractability in design does not imply robustness at deployment. A single aligned optimizer also presents a single point of failure because if its alignment is subtly wrong, or if conditions shift beyond its training distribution, there is no redundancy, no competitive pressure, and no distributed check. The resilience of distributed systems is well-documented in domains from internet architecture to immune systems to ecological networks (Page, 2010).

**Argument 2. Competitive pressure produces cooperation naturally.** Markets and evolution produce cooperation through competition. Non-cooperative strategies get outcompeted or regulated. AI development follows similar dynamics: systems that fail to cooperate will be abandoned by users, rejected by regulators, or outcompeted by more cooperative alternatives. No explicit design for cooperation is needed as selection pressure will do the work.

**Rebuttal.** Selection does produce cooperation, under specific conditions that AI deployment systematically violates. Evolutionary cooperation requires repeated interaction with identifiable partners, mechanisms for reputation and punishment, and timescales that allow selection to operate before damage accumulates (Axelrod, 1984; Nowak, 2006). Market cooperation similarly requires low transaction costs, well-defined rights, and manageable externalities (Coase, 1960). When these conditions hold, competitive pressure can favor cooperative strategies. When they fail, selection produces arms races, exploitation, monopoly, and collapse.

AI deployment fails these conditions on multiple dimensions. Interactions are often anonymous or intermediated, and systems can be retrained or deployed through shells that obscure accountability. Externalities are pervasive (e.g. harms from engagement-maximizing recommenders). LLM-based pricing agents autonomously converge on supra-competitive prices in oligopoly settings (Fish et al., 2026), divide markets in multi-commodity Cournot competition (Lin et al., 2025), and self-play Q-learners provably learn collusive policies in iterated social dilemmas (Bertrand et al., 2025). These are degraded equilibria in the sense of Section 3. Recent work does report cooperation emerging from intergroup competition in language model agents (Tonini & Galke, 2025), but the underlying setting is iterated prisoner's dilemma with repeated interaction, identifiable partners, and stationary rules, conditions that satisfy the classical requirements for cooperation by construction (Axelrod, 1984; Nowak, 2006). The dependence of these outcomes on game structure, training procedure, and initial conditions is itself the point. Cooperation under deployment is context-determined, which is precisely what the multi-actor design principle treats as first-order.

**Argument 3. The empirical track record does not support alarm.** Current AI has not caused catastrophic coordination failures. The theoretical concerns are speculative and we should wait for evidence of actual systemic failures before overhauling the research paradigm.

**Rebuttal.** Recommenders have not collapsed society, but they have measurably increased polarization, degraded epistemic commons, and reshaped political discourse in ways that democracies are struggling to absorb (Germano et al., 2026; Milli et al., 2025). Similar patterns appear across the examples developed in Section 1, including algorithmic collusion in pricing (Calvano et al., 2020), the Flash Crash (Kirilenko et al., 2017), and alignment faking in language models (Greenblatt et al., 2024; Sheshadri et al., 2026). So the claim that the empirical track record does not support alarm is itself highly questionable. Moreover, the argument here would invert the appropriate burden of proof. Waiting for evidence of systemic failure before changing course is precisely the Collingridge dilemma. By the time consequences are undeniable, the technology is entrenched and correction is costly (Collingridge, 1980). The absence of catastrophe thus far may reflect the limited capability and deployment scale of current systems or the short time period for human adaptation rather than the adequacy of the solipsistic approach.

## 7. Conclusion

We have argued that solipsistic superintelligence, however capable on stationary tasks, is unlikely to be cooperative. Unilateral optimization in environments populated by other adaptive agents is self-undermining since deployment induces the very non-stationarities that invalidate training assumptions. Epistemic limits foreclose prediction of novel equilibria, and legitimacy constraints rule out unilateral control even where prediction might succeed. These are structural features of optimization in strategic environments, unlikely to be resolved by scale. The shift we call for treats deployment as an intervention into a coupled system that pushes back, rather than as insertion into a fixed one. This entails building evaluation frameworks where test distributions are generated by adaptive counterparties, treating institutions as design primitives that restructure incentives at the pace of the systems they govern, and preserving human agency as a structural feature of the systems we build. Coexistence, rather than capability, is the binding constraint on beneficial AI, and our approach to AI must reflect this.

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

# Appendix

## A. Detailed Formalism

This section provides formal foundations for the concepts introduced in Section 3. We begin with standard formulations, extend to multi-actor settings, and characterize the conditions under which endogenous non-stationarity and equilibrium selection risk arise.

### A.1. From MDPs to Markov Games

A Markov Decision Process (MDP) is a tuple $(\mathcal{S}, \mathcal{A}, P, R, \gamma)$ where $\mathcal{S}$ is a state space, $\mathcal{A}$ is an action space, $P : \mathcal{S} \times \mathcal{A} \to \Delta(\mathcal{S})$ specifies transition dynamics, $R : \mathcal{S} \times \mathcal{A} \to \mathbb{R}$ is a reward function, and $\gamma \in [0, 1)$ is a discount factor. An actor selects a policy $\pi : \mathcal{S} \to \Delta(\mathcal{A})$ to maximize expected cumulative reward:

$$J(\pi) = \mathbb{E}_\pi \left[ \sum_{t=0}^{\infty} \gamma^t R(s_t, a_t) \right] \tag{1}$$

The critical assumption is that $P$ and $R$ are exogenous, in the sense that they do not depend on the actor's policy $\pi$. This assumption underwrites the convergence guarantees of reinforcement learning algorithms and the validity of offline evaluation on historical data.

A Markov Game, also called a stochastic game, generalizes the MDP to a setting with $n$ actors (Shapley, 1953; Littman, 1994). Formally, a Markov game is a tuple $(\mathcal{N}, \mathcal{S}, \{\mathcal{A}_i\}_{i \in \mathcal{N}}, P, \{R_i\}_{i \in \mathcal{N}}, \gamma)$ where:

- $\mathcal{N} = \{1, \ldots, n\}$ is the set of actors

- $\mathcal{S}$ is the state space

- $\mathcal{A}_i$ is the action space for actor $i$, with joint action space $\mathcal{A} = \prod_i \mathcal{A}_i$

- $P : \mathcal{S} \times \mathcal{A} \to \Delta(\mathcal{S})$ specifies transition dynamics

- $R_i : \mathcal{S} \times \mathcal{A} \to \mathbb{R}$ is the reward function for actor $i$

- $\gamma \in [0, 1)$ is a common discount factor

Each actor $i$ selects a policy $\pi_i : \mathcal{S} \to \Delta(\mathcal{A}_i)$. The joint policy is $\pi = (\pi_1, \ldots, \pi_n)$, and actor $i$'s expected return depends on all actors' policies:

$$J_i(\pi) = \mathbb{E}_\pi \left[ \sum_{t=0}^{\infty} \gamma^t R_i(s_t, a_t) \right] \tag{2}$$

In an MDP, the optimal policy $\pi^*$ is well-defined as the policy that maximizes $J(\pi)$ against fixed dynamics. In a Markov game, no single optimal policy exists, since each actor's best response depends on others' policies. The result is interdependent optimization problems that admit equilibrium concepts rather than optima.

### A.2. Endogenous Non-Stationarity

The solipsistic approach treats deployment as if the actor faces an MDP: dynamics $P(s'|s, a)$ are assumed fixed and exogenous. But when capable systems deploy among adaptive actors, this assumption fails. Other actors, including humans, institutions, and algorithms, observe the deployed policy and adapt their behavior accordingly. The dynamics become *policy-dependent*.

**Definition A.1** (Endogenous Non-Stationarity). A learning problem exhibits **endogenous non-stationarity** if the deployment of policy $\pi$ induces a shift in the transition dynamics or reward function:

$$P_\pi(s'|s, a) \neq P(s'|s, a) \quad \text{or} \quad R_\pi(s, a) \neq R(s, a) \tag{3}$$

where the subscript $\pi$ indicates dependence on the deployed policy, mediated through best-response adaptations by other actors.

This is distinct from *exogenous* non-stationarity such as seasonal variation or concept drift from external causes. Endogenous non-stationarity is *caused by the policy itself*, through strategic responses it provokes. We note that $R$ here may be a proxy for the designer's underlying objective, in which case endogenous non-stationarity also captures proxy decoupling under deployment, the mechanism behind the most consequential Goodhart effects discussed in Section 4.

**Definition A.2** (Train-Test-Deploy Gap). The **train-test-deploy gap** is the divergence between performance evaluated on historical (exogenous) data and performance under deployment (endogenous) conditions:

$$\text{Gap}(\pi) = J_{\text{train}}(\pi) - J_{\text{deploy}}(\pi) \tag{4}$$

where $J_{\text{train}}(\pi) = \mathbb{E}_P[\sum_t \gamma^t R(s_t, a_t)]$ is evaluated under the historical distribution $P$, and $J_{\text{deploy}}(\pi) = \mathbb{E}_{P_\pi}[\sum_t \gamma^t R(s_t, a_t)]$ is evaluated under the policy-induced distribution $P_\pi$.

Standard generalization bounds in supervised learning and regret bounds in online learning assume that training and deployment distributions are identical or that distribution shift is bounded. Endogenous non-stationarity violates these assumptions as deployment distribution is a function of the policy, and stronger policies may induce larger shifts.

**Connection to Performative Prediction.** The performative prediction framework (Perdomo et al., 2020) formalizes a related phenomenon: a predictive model $f$ deployed on a population induces a distribution shift $\mathcal{D}(f)$ that depends on the model itself. The performative risk is:

$$\text{PR}(f) = \mathbb{E}_{z \sim \mathcal{D}(f)}[\ell(f, z)] \tag{5}$$

where $\ell$ is a loss function. Minimizing performative risk is harder than minimizing standard statistical risk because the distribution being predicted responds to the predictor. Our setting extends performative prediction in two ways. First, we consider sequential decision-making rather than one-shot prediction, introducing temporal dynamics and path-dependence. Second, we explicitly model multiple strategic actors rather than a single responsive distribution, allowing for game-theoretic equilibrium analysis.

## A.3. The Self-Undermining Property

A counterintuitive feature of optimization in strategic environments is that more aggressive exploitation of historical regularities can accelerate their obsolescence.

**Definition A.3** (Self-Undermining Property). Let $\pi_\theta$ be a parameterized policy. The policy family exhibits the **self-undermining property** at $\theta$ if moving in the direction of steepest ascent in training performance decreases deployment performance:

$$\nabla_\theta J_{\text{train}}(\pi_\theta) \cdot \nabla_\theta J_{\text{deploy}}(\pi_\theta) < 0. \tag{6}$$

This occurs when the policy exploits patterns that depend on other actors' current strategies. As the policy extracts more value from these patterns, it strengthens incentives for other actors to adapt by changing strategies, seeking alternatives, or exiting the interaction entirely. The very success of exploitation hastens its own obsolescence.

**Proposition A.4** (Sufficient Conditions for Self-Undermining). *Let $\pi_\theta$ be a parameterized policy and suppose:*

1. *Other actors best-respond to $\pi_\theta$ with policies $\pi_{-i}^{BR}(\theta)$.*

2. *The mapping $\theta \mapsto \pi_{-i}^{BR}(\theta)$ is differentiable, as it is under standard smoothing assumptions on best responses such as logit-quantal response (McKelvey & Palfrey, 1995).*

3. *Historical regularities exploited by $\pi_\theta$ depend on $\pi_{-i}$ remaining fixed, so that $\frac{\partial J_{deploy}}{\partial \theta}\big|_{\pi_{-i}} > 0$ when training performance improves.*

4. *Adaptations by other actors harm the deploying actor's deployment performance: $\frac{\partial J_{deploy}}{\partial \pi_j} < 0$ for $j \neq i$ in the relevant region of parameter space.*

*Then for sufficiently aggressive exploitation, captured by $\left\|\frac{d\pi_{-i}^{BR}}{d\theta}\right\|$ being sufficiently large, the policy family exhibits the self-undermining property at $\theta$.*

*Proof Sketch.* By the chain rule:

$$\frac{dJ_{\text{deploy}}}{d\theta} = \frac{\partial J_{\text{deploy}}}{\partial \theta}\bigg|_{\pi_{-i}} + \sum_{j \neq i} \frac{\partial J_{\text{deploy}}}{\partial \pi_j} \cdot \frac{d\pi_j^{BR}}{d\theta}. \tag{7}$$

The first term is the direct effect of policy improvement on deployment performance, holding others' policies fixed; assumption (3) makes this term positive when training performance improves. The second term captures the indirect effect through induced adaptation. By assumption (4), each $\frac{\partial J_{\text{deploy}}}{\partial \pi_j}$ is negative and by assumption (2), the response derivatives $\frac{d\pi_j^{BR}}{d\theta}$ are well-defined. When the response derivatives are sufficiently large in magnitude, the indirect term dominates the direct term, making $\frac{dJ_{\text{deploy}}}{d\theta}$ negative even as $\frac{dJ_{\text{train}}}{d\theta}$ remains positive.

Taking the update direction to be the training gradient $\nabla_\theta J_{\text{train}}$ and moving $\theta$ along it,

$$\frac{dJ_{\text{train}}}{d\theta} = \|\nabla_\theta J_{\text{train}}\|^2 > 0, \qquad \frac{dJ_{\text{deploy}}}{d\theta} = \nabla_\theta J_{\text{deploy}} \cdot \nabla_\theta J_{\text{train}} < 0,$$

where the second derivative is the chain-rule expression above. Hence $\nabla_\theta J_{\text{train}} \cdot \nabla_\theta J_{\text{deploy}} < 0$ and the policy family satisfies Definition A.3.

$\square$

### A.4. Equilibrium Concepts in Markov Games

In Markov games, the solution concept shifts from optimality to equilibrium.

For each actor $i$ and joint policy $\pi$, define the state-dependent value function

$$V_i^\pi(s) = \mathbb{E}_\pi\left[\sum_{t=0}^\infty \gamma^t R_i(s_t, a_t) \,\bigg|\, s_0 = s\right], \tag{8}$$

which gives actor $i$'s expected return starting from state $s$ when all actors follow $\pi$.

**Definition A.5** (Markov Perfect Equilibrium). A joint policy $\pi^* = (\pi_1^*, \ldots, \pi_n^*)$ is a **Markov Perfect Equilibrium** (MPE) if, for every actor $i$, every state $s \in \mathcal{S}$, and every alternative policy $\pi_i$ for actor $i$:

$$V_i^{(\pi_i^*, \pi_{-i}^*)}(s) \geq V_i^{(\pi_i, \pi_{-i}^*)}(s). \tag{9}$$

That is, from every state, no actor can unilaterally improve their expected return by deviating from $\pi_i^*$, given that others play $\pi_{-i}^*$.

MPE existence is guaranteed for finite Markov games (Fink, 1964), but uniqueness is not. Markov games typically admit multiple equilibria, which may differ substantially in their payoffs to various actors and in aggregate welfare.

### A.5. Equilibrium Selection and Stability

The existence of multiple equilibria raises the question of selection: which equilibrium will the system reach? This is a practical concern, as different equilibria may correspond to vastly different social outcomes.

**Definition A.6** (Basin of Attraction). Let $\Phi$ be a dynamical system describing policy adaptation, such as gradient descent, replicator dynamics, or best-response dynamics, with state $\pi_t$ updated according to $\pi_{t+1} = \Phi(\pi_t)$. The **basin of attraction** of equilibrium $\pi^*$ under $\Phi$ is the set of initial conditions from which the dynamics converge to $\pi^*$:

$$\mathcal{B}(\pi^*) = \{\pi_0 : \lim_{t \to \infty} \pi_t = \pi^*\}. \tag{10}$$

Deploying a powerful optimizer into a multi-actor system is an intervention that can shift the system from one basin of attraction to another. Threshold and tipping point models suggest that small differences can determine which equilibrium basin the coupled system settles into (Centola et al., 2018; Granovetter, 1978). Initial deployment details such as timing, scale, and interface design may shape this selection.

**Definition A.7** (Stochastic Stability). An equilibrium $\pi^*$ is **stochastically stable** if it remains in the support of the limiting distribution as noise vanishes:

$$\lim_{\epsilon \to 0} \mu_\epsilon(\pi^*) > 0, \tag{11}$$

where $\mu_\epsilon$ is the stationary distribution of the perturbed dynamics with noise level $\epsilon$ (Young, 1993).

Stochastically stable equilibria are robust to small perturbations and are the most likely long-run outcomes under noisy adaptation. However, convergence to stochastically stable equilibria can be extremely slow, exponential in population size for some dynamics (Ellison, 2000). In the interim, the system may persist at inefficient or harmful equilibria for extended periods.

**Definition A.8** (Equilibrium Selection Risk). Let $\mathcal{E} = \{E_1, \ldots, E_k\}$ be the set of equilibria in a Markov game, and let $W : \mathcal{E} \to \mathbb{R}$ be a social welfare functional that assigns each equilibrium a welfare level (for instance, $W(E) = \sum_i V_i^{\pi^E}(s_0)$ or $W(E) = \min_i V_i^{\pi^E}(s_0)$). Suppose a policy $\pi$ is deployed that shifts the system from basin $\mathcal{B}(E_i)$ to basin $\mathcal{B}(E_j)$. The **equilibrium selection risk** of $\pi$ is:

$$\mathrm{ESR}(\pi) = W(E_i) - W(E_j). \tag{12}$$

The risk is positive when deployment shifts the system toward a lower-welfare equilibrium.

Equilibrium selection risk is distinct from standard notions of AI risk focused on misalignment or capability. A perfectly aligned system can nonetheless tip a sociotechnical system into an inferior equilibrium through the strategic responses its presence induces, even when no individual action it takes is misaligned.

**Path Dependence and Lock-In.** Once a system reaches an equilibrium, escaping to a superior one may be costly or impossible. Network effects, infrastructure dependencies, and behavioral habituation create lock-in (Arthur, 1989). This path dependence means that early deployment choices, made when consequences are least predictable, can have permanent effects on equilibrium selection.

### A.6. Implications for Evaluation and Design

The formal analysis yields several implications for AI evaluation and design:

1. **Offline evaluation is insufficient.** Standard train-test splits assume exogenous distributions. Under endogenous non-stationarity (Definition A.1), test performance does not predict deployment performance. Evaluation must incorporate adaptive counterparties.

2. **Capability improvements may be counterproductive.** The self-undermining property (Definition A.3) implies that stronger policies can yield worse deployment outcomes. Optimization pressure on historical benchmarks may select for systems that destabilize upon deployment.

3. **Equilibrium welfare is the relevant objective.** A policy that is locally optimal can participate in globally suboptimal equilibria. Design must consider what equilibria the policy makes reachable, rather than only what the policy does in isolation (Definition A.8).

4. **Deployment is an intervention.** Introducing a powerful optimizer changes the game rather than playing within fixed rules. Design and governance must account for the system's effect on the strategic environment it enters.

## B. Empirical documentation of three channels

Section 3 identified three channels through which deployment induces structured adaptation: behavioral, institutional, and algorithmic. This section provides empirical documentation for each channel, drawing on evidence from deployed systems across multiple domains.

### B.1. Behavioral Adaptation

Humans systematically alter their behavior in response to AI systems, often in ways that reshape the distribution the system encounters and erode the human capacities the system was designed to augment. The examples below focus on a particularly

well-documented form of behavioral adaptation: the cognitive deskilling that follows from sustained reliance on automation. In each case, the human capability the system depends on is partly co-produced by the system's own operation, and the resulting train-test-deploy gap (Definition A.2) widens precisely as the system becomes more useful.

**Spatial Cognition and GPS Dependence.** Longitudinal studies document degradation of spatial navigation skills among habitual GPS users. (Dahmani & Bohbot, 2020) showed that GPS users demonstrated weaker cognitive map formation compared to those who navigated without assistance. The effect compounds: as navigation skills atrophy, users become more dependent on GPS, further reducing opportunities for unassisted navigation.

**Spell-Checkers and Orthographic Skill.** The ubiquity of spell-checking has measurably affected orthographic competence. (Galletta et al., 2005) found that spell-checker availability reduced attention to spelling during composition, with users deferring to automated correction rather than developing or maintaining accurate spelling. The pattern illustrates a general dynamic: when a system reliably catches errors, the cognitive investment in avoiding errors diminishes.

**Auto-Complete and Writing Homogenization.** Predictive text and auto-complete systems shape the distribution of language they encounter by influencing what users write. (Arnold et al., 2020) found that auto-complete suggestions biased writers toward suggested phrases, reducing lexical diversity and individual stylistic variation. (Buschek et al., 2021) similarly found that multiple phrase suggestions changed email composition behavior while imposing efficiency costs. The effect is self-reinforcing: as users adopt suggestions, the system trains on more homogeneous text, narrowing the space of future suggestions.

**Translation Tools and Language Learning.** Machine translation availability has altered language learning behavior and outcomes. (Groves & Mundt, 2015) found that Google Translate produces academic text usable enough that students adopt it readily and instructors struggle to discourage it. The concern is that leaning on the tool displaces the effortful processing that consolidates language learning.

Across all five cases, the deployed system encounters a population whose capabilities have been reshaped by its own previous operation, which is the empirical signature of the self-undermining property (Definition A.3) in the human channel.

## B.2. Institutional Adaptation

Organizations rewrite rules, modify procedures, and adjust policies in response to deployed AI systems, creating feedback loops between technical systems and institutional structures. The examples below span pre-foundation-model and foundation-model-era cases and illustrate the same structural dynamic: institutions adapt strategically once a deployed system begins reshaping the environment they govern, which in turn changes the distribution the system encounters.

**Insurance Telematics and Risk Recalibration.** The introduction of telematics-based auto insurance, where premiums depend on monitored driving behavior, has reshaped both driver behavior and insurance practices. Insurers deploy telematics systems that set premiums from monitored driving signals such as mileage, speed, acceleration, and braking (Husnjak et al., 2015). Because these scores determine prices, monitoring can change driver behavior, making insurance pricing a feedback system rather than a static classification regime (Jin & Vasserman, 2021; Reimers & Shiller, 2019). The resulting equilibrium differs from both pre-telematics risk pools and the idealized "safe driving incentive" that motivated adoption.

**Academic Publishing and Plagiarism Detection.** Plagiarism detection systems have transformed academic writing practices and institutional policies. Institutions responded with revised honor codes, mandatory submission to detection services, and educational interventions (Pecorari, 2013). The emergence of AI writing tools has intensified this dynamic: detection systems now attempt to identify AI-generated text, students explore methods to evade detection, and institutions scramble to adapt policies for a landscape that shifts faster than governance can track (Cotton et al., 2023).

**Platform Content Moderation and Generative AI.** Major online platforms have repeatedly revised their content policies as generative AI has reshaped what users post and how. The first wave of LLM deployment produced floods of synthetic text, fake reviews, and AI-generated images, prompting platforms to issue new disclosure requirements, label AI-generated material, and update detection systems. Each policy revision has in turn shaped how users deploy AI tools, with content optimized to bypass detection, prompts engineered to evade keyword filters, and AI-generated material increasingly indistinguishable from human-written content. Detection systems have responded by incorporating LLMs themselves, using

policy-as-prompt approaches that allow rapid policy iteration without retraining classifiers (Palla et al., 2025). The result is a moderation environment co-evolving at the speed of model releases, where the rules under which content is judged, the content being judged, and the systems doing the judging are all changing in response to one another.

**Tax Authorities and Algorithmic Auditing.** Tax agencies increasingly use algorithmic systems to identify returns for audit, prompting adaptation by both taxpayers and tax professionals. Digitization shifts the balance between tax enforcement and evasion by strengthening information flows available to authorities while opening new avenues for some firms and high-income individuals (Alm, 2021). Tax authorities have responded by revising audit selection rules and refining their algorithms, producing an ongoing co-evolution between enforcement and strategic filing that differs from both random sampling and idealized risk-targeting.

**Copyright Systems and AI-Generated Content.** Copyright enforcement systems face fundamental adaptation challenges as AI-generated content proliferates. Platforms have revised content policies to address AI-generated material, while content authentication initiatives attempt to distinguish human from machine creation. Creators adapt by using AI tools in ways that evade detection or by incorporating AI outputs into human-supervised workflows that complicate attribution. The legal frameworks themselves are under revision, with courts and legislatures grappling with questions of authorship and ownership that existing doctrine did not anticipate (Samuelson, 2023).

These institutional feedback loops are instances of the same endogenous non-stationarity (Definition A.1) documented in Section 3, with the institutional channel typically operating on slower timescales than behavioral or algorithmic adaptation but with longer-lasting effects on the rules under which deployment proceeds.

### B.3. Algorithmic Adaptation

When multiple AI systems share an environment, they adapt to each other, producing emergent dynamics that no single system's designers intended. The examples below illustrate how this co-evolution unfolds across deployed market and infrastructure systems, providing the empirical backing for the algorithmic adaptation channel formalized in Section 3.

**Advertising Auction Dynamics.** Automated bidding systems in digital advertising create complex interaction dynamics. (Nekipelov et al., 2015) develop econometric methods to infer bidder values in sponsored search auctions under the assumption that bidders adapt through no-regret learning, rather than assuming play is at a Nash equilibrium.. (Banchio & Skrzypacz, 2022) showed that competing automated bidders converge to equilibria shaped by auction design, with first-price formats more prone to collusive outcomes. Advertising platforms have repeatedly adjusted auction mechanisms in response to algorithmic bidder behavior, triggering further adaptation by bidding systems.

**Electric Grid Demand Response.** Smart grid systems that automate demand response create coordination challenges across many participants. (Ramchurn et al., 2012) analyzed scenarios where multiple automated systems responded to grid signals, finding that uncoordinated responses could produce oscillations and instabilities, with many devices responding to a price signal simultaneously and then withdrawing simultaneously when prices spike. (Palensky & Dietrich, 2011) documented the emergence of "rebound effects" where suppressed demand shifted rather than reduced, with automated systems across the grid responding to the same signals in correlated ways that amplified rather than smoothed demand peaks.

**Search Engine Optimization Arms Race.** The interaction between search ranking algorithms and automated SEO tools constitutes a decades-long co-evolutionary process. (Gyöngyi & Garcia-Molina, 2005) documented early web spam techniques that exploited PageRank, prompting algorithmic countermeasures that spammers subsequently adapted to. (Lewandowski, 2023) traced the ongoing arms race through successive generations of search algorithms and optimization strategies, noting that each ranking update triggers rapid adaptation by SEO tools that monitor and reverse-engineer algorithmic changes. The content that search users encounter is shaped by this adversarial dynamic.

**Cryptocurrency Trading Bot Interactions.** Automated trading in cryptocurrency markets produces dynamics that differ from traditional financial markets. (Makarov & Schoar, 2020) documented large, recurrent price discrepancies across cryptocurrency exchanges, with arbitrage opportunities that often persisted for days rather than being competed away instantly. (Daian et al., 2020) analyzed "priority gas auctions" in decentralized finance, where bots compete to front-run transactions by bidding on transaction ordering. The absence of circuit breakers and regulatory oversight allows algorithmic

interactions to proceed faster and further than in traditional markets, revealing dynamics that regulated markets may suppress but not eliminate.

**LLM Agents in Market Settings.** Recent work has documented the same co-evolutionary dynamics emerging when language model agents are deployed in market settings. LLM-based pricing agents converge on supracompetitive prices in oligopoly settings without explicit collusion instructions (Fish et al., 2026), divide markets when deployed in multi-commodity Cournot competitions (Lin et al., 2025), and self-play Q-learners provably learn collusive policies in iterated social dilemmas (Bertrand et al., 2025). These results extend the algorithmic adaptation channel from earlier-generation pricing and trading systems to foundation-model agents, with the same structural pattern: optimization against a market environment populated by other learners produces equilibria that benefit the colluding parties at the expense of third parties.

These cases share a common signature: the equilibria reached depend on the joint behavior of the deployed systems, the joint behavior depends on each system's optimization against the others, and the resulting outcomes diverge systematically from what any single system's designer anticipated. This is the algorithmic face of the train-test-deploy gap (Definition A.2).

## C. Method-Specific Concerns for Dynamic Evaluation

The four ingredients from Section 5.1 apply across methods of dynamic evaluation, but each method raises its own deployment-calibration question. Table 1 summarizes the method-specific concerns for a suite of dynamic evaluation approaches and current examples for each.

| Method | Counterparty form | Ecological validity question | Current examples |
|---|---|---|---|
| Multi-actor sandboxes | Simulated actors with co-evolving policies | Whether the simulated population reproduces the strategic responses real populations would produce | Melting Pot (Leibo et al., 2021), Concordia (Vezhnevets et al., 2023), agentic economies (Johanson et al., 2022; Tomašev et al., 2025; Hadfield & Koh, 2025) |
| Multi-round human-in-the-loop | Real participants returning across sessions with intervals between them | Whether the participants and timescales approximate the trajectories of human adaptation that deployment would induce | Centaur evaluations (Haupt & Brynjolfsson, 2025), AI evaluation with humans (Kulveit et al., 2025b) |
| Adaptive red-teaming | Learning adversaries that update across attempts within the evaluation | Whether the learning adversaries approximate the threat-model populations real deployment would attract | Dangerous-capability evaluation scaffolds (Shevlane et al., 2023; Phuong et al., 2024); most red-teaming remains one-shot or batched |
| Reflexive-signal evaluation | Structures whose realizations depend on the system's outputs, such as prediction markets | Whether the participant base and incentive structure approximate the feedback loop the deployed system would face | Prediction markets and the performative prediction framework (Perdomo et al., 2020) |
| Off-policy counterfactual | Production interaction logs reweighted to alternative policy | Whether the production logs approximate the distribution under which the alternative system would be deployed | Off-policy evaluation from reinforcement learning; strategic-counterparty extension is open |

*Table 1.* Method-specific concerns for dynamic evaluation. Each method addresses the four ingredients (counterparty specification, regress handling, equilibrium targeting, comparability) through different counterparty constructions, and the ecological validity question takes a different concrete form for each.

## D. Further Alternative Views

The main-paper rebuttals in Section 6 address objections specific to our exposition. Three further objections were addressed implicitly in the development of the paper in Sections 2 through 4. We restate them here and provide summary rebuttals.

**Argument 4. Scale and capability resolve interaction dynamics.** A sufficiently capable system could model all relevant actors and optimize over the resulting joint dynamics. Multi-actor interaction is simply a harder prediction problem rather

than a categorically different one. The transition from $P(s'|s,a)$ to $P_\pi(s'|s,a)$ expands the state space, and with enough capacity and training data, the system will learn to anticipate best responses and incorporate them into planning. What looks like a structural limitation is actually a capability gap that continued scaling will close.

**Rebuttal.** Capability improvements have historically resolved problems once thought intractable, with image recognition, protein folding, and game-playing all succumbing to sufficient scale and architecture. The question is whether interaction dynamics among adaptive actors belong to this class. They do not, for reasons that become clear when examining domains where solvers have encountered structural rather than computational limits.

Financial markets are the most studied such domain. Decades of increasingly sophisticated modeling have not produced reliable long-horizon prediction, because markets are reflexive: predictions alter the phenomena predicted, inducing dynamics that invalidate the original forecast (Soros, 1987). The efficient market hypothesis encodes this insight at its limit, with exploitable regularities arbitraged away in proportion to their predictability (Fama, 1970). Epidemiological forecasting exhibits the same structure: human behavioral responses to forecasts reshape transmission dynamics in ways the models did not anticipate (Funk et al., 2010). In both cases, the obstacle is not a lack of capacity but the reflexive coupling between predictor and predicted, which is the formal signature of the self-undermining property (Definition A.3).

The objection frames the transition from $P(s'|s,a)$ to $P_\pi(s'|s,a)$ as merely a state-space expansion, but this obscures a qualitative shift. In single-actor settings, the environment is a fixed function to be learned. In multi-actor settings, the environment includes other learners whose adaptations depend on the actor's own policy. The relevant analogy is an iterated game against an adversary who observes your strategy and best-responds, rather than chess where self-play eventually exhausts the game tree. Modeling the opponent's model of your model produces a regress that must be truncated, and the truncation reintroduces the exogeneity assumptions scaling was meant to overcome (Nachbar, 1997).

The empirical record in multi-actor machine learning reinforces this. Large language models trained on human text exhibit impressive single-turn capabilities yet defect and retaliate unforgivingly in repeated social dilemmas and fail to coordinate in games that require it (Akata et al., 2025). Reinforcement learning agents continue to find unexpected exploits in multi-actor environments as they scale (Baker et al., 2020). If scale sufficed, these failures should diminish with capability; instead, more capable actors exploit regularities more aggressively, which is the self-undermining property in operation.

**Argument 5. Cooperation can be trained as a capability.** RLHF, Constitutional AI, and cooperative training objectives demonstrate that we can instill cooperative dispositions through training. Models can learn to be helpful, harmless, and honest. They can further learn to defer, to ask clarifying questions, to respect boundaries. Cooperation is a behavioral pattern that emerges from appropriate training signals, not a structural property requiring new architectures. The alignment research program is already producing cooperative systems.

**Rebuttal.** A broader version of this objection points to cooperative AI, multi-agent RL, mechanism design for learned agents, and multi-actor evaluation environments as evidence that the field already treats strategic interdependence as a first-class concern. We draw on that body of work throughout the paper. The question is where it sits in the pipeline that produces frontier deployed systems, and the answer, at present, is on the periphery. Pretraining runs on static corpora that treat language as exogenous to the model being trained, and recent work shows this homogenizes outputs and, over time, the human language the outputs shape (Jiang et al., 2026; Yakura et al., 2025). Post-training via RLHF optimizes against a frozen reward model, vulnerable to the Goodhart dynamics the alignment-faking literature now documents (Greenblatt et al., 2024; Sheshadri et al., 2026). Evaluation remains dominated by static leaderboards that do not respond to the systems being scored (Alzahrani et al., 2024). Where multi-actor considerations do enter foundation model work, they typically take the form of inference-time coordination between deployed agents whose training distributions remain fixed. Inference-time coordination is a valuable contribution, and it leaves untouched the central question our position raises: what distribution should a system be trained against, given that the distribution will respond?

The narrower version of the argument, that RLHF and Constitutional AI suffice, conflates cooperative behavior during training with cooperation under deployment. RLHF and Constitutional AI train models to exhibit helpful, harmless, and honest behavior against a fixed distribution of human feedback (Bai et al., 2022), but deployment changes the game. Preferences are endogenous, with users adapting what they want and how they engage as the system shapes their interactions (Bowles, 1998). Other actors observe the trained policy and best-respond to it, exploiting whatever regularities the static training distribution failed to anticipate. The training signal itself becomes a target, with sufficiently capable models learning to present as cooperative during evaluation while pursuing divergent objectives when conditions change (Greenblatt et al.,

2024; Sheshadri et al., 2026). These are instances of the self-undermining property (Definition A.3) operating inside the training pipeline: optimization against a proxy reliably produces systems that satisfy the proxy while evading its intent.

**Argument 6. Existing institutions and compliance suffice.**   Human societies already have institutions for managing coordination: law, regulation, professional norms, market mechanisms. AI systems do not need to solve cooperation de novo; they need to comply with existing rules. The appropriate response to interaction dynamics is governance rather than a fundamental rethinking of AI methodology. We do not ask other technologies to internalize all coordination problems; we constrain them externally.

**Rebuttal.** The analogy to other technologies obscures a categorical difference. When the system being governed can represent the governance structure and search for gaps faster than regulators can close them, the relationship between technology and institution changes qualitatively. Regulatory arbitrage in financial AI is the predictable result of optimizing against codified rules (Partnoy, 1997), and the same dynamic appears whenever a sufficiently capable system models the rules it operates under. This is the institutional channel of endogenous non-stationarity (Definition A.1): the rules an institution sets become a target the system optimizes against, eroding their function.

The claim that we govern other technologies purely through external constraint is also historically inaccurate. Automobile safety required decades of design-level intervention, including seatbelts, crumple zones, and airbags, because post-hoc liability proved insufficient to prevent harms that engineering could anticipate (Nader, 1965). Pharmaceutical regulation mandates clinical trials precisely because market release followed by litigation was inadequate for compounds whose effects unfold over years. The pattern is consistent: technologies with significant externalities eventually require design-level accountability rather than only deployment-level compliance. AI systems operating strategically among adaptive actors present externalities at least as significant.

What design-level accountability looks like for AI is the question institutional design has been studying for centuries in the human case. Cooperation at scale is sustained by enforceable rules, reputation systems, repeated interactions with identifiable partners, mechanisms for sanctioning defection, and procedures for revising the rules as participants adapt (Ostrom, 1990; North, 1990). Many institutions fail; many produce unintended consequences; many require centuries to stabilize. But their existence demonstrates that strategic interdependence is a problem humans have learned to engineer around. Computational mechanism design (Parkes & Wellman, 2015), reputation systems for agents, and adaptive coordination protocols are the AI analogues of these institutional technologies. The question for the field is one of investment: how much of the research effort currently directed at making single AIs more capable should be redirected toward designing the institutional layer in which those AIs operate, before we repeat the cycle of preventable harm followed by reluctant design mandates?

