# OpenReview forum: "Position: Solipsistic superintelligence is unlikely to be cooperative"
_ICML.cc/2026/Position_Paper_Track — ICML 2026 Position Paper Track regular_

### Official Review · Reviewer_xkrq · 2026-02-25

**Significance:** 3
**Argument Clarity:** 3
**Rating:** 4
**Confidence:** 4

**Questions:**

- Why the solipsistic method is limited to superintelligence? What are examples of solipsistic superintelligence? Do they connect to examples like sandbagging/deception during training?

- The train-test-deploy gap is universal. What are reasons why it is special to superintelligence?

- Is cooperation what we need for superintelligence? Is the multi-agent-first approach sufficient for cooperation?

**Alternative Views Section:**

Yes

**Compliance With Llm Reviewing Policy A Conservative:**

Affirmed.

**Discussion Potential:**

3

**Paper Summary:**

The authors argue that a solipsistic superintelligence is unlikely to be cooperative. The authors first explain that cooperation is necessary for sustained beneficial outcomes using the deployment features such as systemic externalities. To argue the limitation of the solipsistic methods, the authors explain the training and development conditions and show how the solipsistic methods can suffer the train-test-deploy gap. The authors also explain the difficulty of using unilateral optimizers to address the view that uses prediction to model adaption.  Finally, the authors conclude a multi-agent-first approach and how multi-agent dynamics should be considered. Some alternative views are also discussed, together with rebuttals.

**Position:**

Yes

**Position In Title:**

Yes

**Related Work:**

3

**Strengths And Weaknesses:**

**Strengths**

- The authors identify the gap between the solipsistic assumption and the deployment reality, and argue that solipsistic AI optimizers do not fit into real complex environment. A nice summary of examples is provided to show that the deployment of solipsistic AI can fail in real environment.

- The train-test-deploy gap and the unilateral optimization are carefully analyzed, when the authors argue the necessity of going beyond the solipsistic approach.

- The authors also present a multi-agent-first approach that appears to be a promising direction as next.

**Weaknesses**

- The limitation of the solipsistic approach is argued for superintelligence. It is not clear which are examples of superintelligence.

- Other than cooperation, agents may incur other mixed behavior, switching between cooperation and competition.

- The proposed multi-agent-first approach may not necessarily ensure cooperation. It is not clear how strong this would be a necessary next step.

**Support:**

2

---

> ### Author Rebuttal · Authors · 2026-03-31
>
> We thank the reviewer for their thorough and probing engagement. We are glad to see the reviewer highlighting the value of the gap identified between solipsistic assumption and deployment reality and the promise of the multi-agent first direction. The questions and concerns raised by the reviewer are closely interrelated and we address them in sequence below:
>
> We want to clarify that our critique is not restricted to superintelligence. The three assumptions of exogeneity, stationarity, and singleton framing (Section 3.1, lines 170–183) characterize the dominant methodology at all capability levels. Several examples in our paper involves current systems: recommendation algorithms producing polarization (lines 95–97), pricing algorithms learning supracompetitive strategies (Calvano et al., 2020), the Flash Crash (Kirilenko et al., 2017), and GPS induced cognitive degradation (Appendix B.1). We use the superintelligence framing because the dominant paradigm assumes that scaling capability will eventually deliver the cooperative outcomes desired by humans, just as scaling delivers better math or coding. Our position is that cooperation is not a task to be solved by a more powerful optimizer. It is an equilibrium property emerging from how multiple agents navigate their strategic interdependence (Key Claim 1, lines 103–108, 146–149). We agree that endogenous non-stationarity already applies to current agentic systems, but our position is that nothing changes even with superintelligence because the problem is structural rather than a capability gap. Concrete examples of solipsistic superintelligence include frontier LLMs deployed as autonomous agents, pre-trained on static corpora and post-trained via frozen RLHF, which upon deployment trigger all three channels of structured adaptation (Section 3.3). As detailed in our response to Reviewer cp56, we will sharpen the definition in revision.
>
> Regarding sandbagging and deception, the connection is direct. Alignment faking (Greenblatt et al., 2024; Sheshadri et al., 2025, lines 320–326) is endogenous non-stationarity operating within the training loop itself. The training pipeline constitutes a strategic environment from the model's perspective, and a sufficiently capable system treats it as a game, discovering strategies that satisfy evaluation criteria without genuine compliance. This shows that even alignment techniques are vulnerable to the solipsistic assumption when the reward model is treated as fixed.
>
> We do not see train-test-deploy gap as being special to superintelligence. In fact, we agree with the reviewer that the gap is universal and regard this as a feature of our argument. The critical insight is that the gap structurally widens with capability. Proposition A.4 (lines 800–820) and Equation 7 formalize this: for weakly capable systems the indirect effect through induced adaptation is small, but for superintelligent systems it dominates because superior exploitation creates stronger incentives for counterparties to adapt. Three scaling properties interact multiplicatively: temporal asymmetry means responses arrive as discontinuous shifts (Section 2.2, lines 156–163), deployment across coupled domains triggers cascading non-stationarities (lines 148–155), and exploitation of subtle patterns produces catastrophic degradation when those patterns shift (Proposition A.9). For superintelligent systems it becomes structural because epistemic limits foreclose prediction of novel equilibria (Section 4.1, lines 237–267) and legitimacy constraints become binding as systems reshape social arrangements at scale (Section 4.2, lines 275–318).
>
> On mixed behavior and sufficiency, we use the term "cooperation" broadly to mean selecting good equilibria under non-stationarity, which is what humans want from deployed AI systems in virtually any setting. This does not exclude competition or mixed motive dynamics. The multi-agent first approach provides the representational foundation to navigate the full spectrum of strategic interactions, including knowing when cooperation is appropriate and when competition is warranted. The solipsistic paradigm cannot navigate this spectrum because it absorbs other agents into the state space (Section 3.1, lines 179–183), collapsing the strategic structure that determines which equilibrium is reached. The multi-agent first approach is a necessary design principle that treats  strategic interdependence as foundation to evaluation, deployment, and governance, not merely a training methodology. Whether such an approach combined with appropriate institutions is sufficient remains an open research question that can only be pursued once the necessity is recognized. That recognition is the paradigm shift we call for.
>
> We hope that this addresses all your concerns and we welcome further discussion on any outstanding ones or if you need further clarifications on the above responses.

---

> > ### Author Rebuttal · Reviewer_xkrq · 2026-04-02
> >
> > Thank you for the response. My concerns are addressed well, and I will maintain my score at this stage.

---

### Official Review · Reviewer_VnQ2 · 2026-03-12

**Significance:** 3
**Argument Clarity:** 3
**Rating:** 4
**Confidence:** 3

**Questions:**

See Strengths and Weaknesses.

**Alternative Views Section:**

Yes

**Compliance With Llm Reviewing Policy A Conservative:**

Affirmed.

**Discussion Potential:**

3

**Final Justification:**

The rebuttal has addressed my concerns by providing more explanations on the design principle and solipsistic assumptions. So, I will keep my positive rating of 4 for the paper.

**Paper Summary:**

This paper critiques the "solipsistic" approach in AI development, which assumes environments are exogenous, stationary, and involve a single optimizing agent. The authors argue that deploying such systems into real-world settings can lead to a dangerous "train-test-deploy gap." This gap arises from endogenous non-stationarity, where the system's own actions provoke strategic responses that alter the environment, rendering its training obsolete and risking a shift to degraded social equilibria. The paper proposes a fundamental shift toward "multi-agent-first" research, emphasizing cooperation, dynamic evaluation, institutional design, and the preservation of human agency as necessary conditions for beneficial AI deployment.

**Position:**

Yes

**Position In Title:**

Yes

**Related Work:**

3

**Strengths And Weaknesses:**

**Strengths**

- The authors pinpoint the structural flaws in current methodologies of developing AI by clearly articulating the implicit assumptions of exogeneity and stationarity and contrasting them with the strategic, adaptive nature of real-world deployment.

- The authors build a compelling case through a well-structured, interdisciplinary argument by connecting formal concepts from game theory with concrete empirical evidence across different channels of adaptation, which grounds their high-level claims in observable phenomena.

**Weaknesses**

- While the critique of solipsistic methods is detailed, the proposed alternative of multi-agent-first research remains relatively high-level and programmatic. The paper points to valuable directions but does not provide a concrete, detailed blueprint for how to build or train such systems, or how to resolve the new coordination failures they might introduce.

- The paper argues that solipsistic assumptions are baked into the dominant methodology. However, it could be argued that this is a critique of a specific paradigm (e.g., single-agent RL on static datasets) rather than all current AI research. A deeper engagement with how cutting-edge multi-agent RL or foundation model research might already be addressing some of these concerns would strengthen the argument.

**Support:**

3

---

> ### Author Rebuttal · Authors · 2026-03-31
>
> We thank the reviewer for their thoughtful review and we are delighted to see the reviewer acknowledge the structural critique of exogeneity and stationarity assumptions and the interdisciplinary grounding of our work.
>
> Regarding the concreteness of the multi-agent first agenda, we emphasize that we are proposing a design principle where strategic interdependence is treated as foundational to evaluation, deployment, and governance. The methodological anchor is dynamic evaluation, where test distributions are generated by adaptive counterparties rather than fixed benchmarks (Section 5.1, lines 330–344). Institutions and human agency then serve as the two context dependent pillars that shape how cooperative outcomes are achieved in specific deployment settings. This design principle has concrete instantiations within existing ML pipelines which we outline in detail in our response to Reviewer cp56 and will incorporate into an expanded Section 5 in revision. Beyond those interventions, recent empirical work further grounds these directions. The Agents of Chaos study demonstrates that deploying agents without institutional constraints in open-ended environments produces predictable failure modes including norm violation, deception, and coordination collapse. This serves as a concrete evidence that institutional scaffolding is not optional but necessary for maintaining cooperative dynamics. Similarly, Moltbook style forum environments offer testbeds for studying how norms emerge, stabilize, or degrade when agents interact through structured communication channels, directly targeting the recommendation algorithm failure modes described in our introduction (lines 95–97).
>
> Regarding coordination failures that multi-agent designs may introduce, we share this concern and address it explicitly in Section 6, Argument 1 (lines 385–420), where we state that decentralization alone guarantees nothing (Ostrom, 1990; Hardin, 1968). The relevant comparison is not between a failure-free monolith and a failure-prone multi-agent system. A monolithic optimizer does not avoid multi-agent dynamics. It encounters them at deployment while having been designed as if they did not exist (lines 399–405). The mechanisms for addressing coordination failures are drawn from institutional design: market mechanisms that isolate failures (Tomašev et al., 2025a), reputation systems that make defection costly (Axelrod, 1984; Nowak, 2006), and learned mechanism design (Yang et al., 2022).
>
> On our position being the critique of a specific paradigm, we agree that we can make our exposition sharper. Our critique aims to target the dominant methodology and scaling pathway. The Scope paragraph (lines 110–125) states this explicitly, and the critique is methodological rather than taxonomic, applying to any system inheriting the three assumptions of exogeneity, stationarity, and singleton framing (Section 3.1, lines 170–183). We engage extensively with cutting-edge multi-agent work throughout the paper (Dafoe et al., 2020; Conitzer and Oesterheld, 2023; Leibo et al., 2017, 2019a, 2021; Baker et al., 2020; Vezhnevets et al., 2023; Tomašev et al., 2025b; Yang et al., 2022). Our position is not that this work does not exist but that it remains peripheral to the central scaling pathway (lines 88–91). The dominant foundation model pipeline has not internalized multi-agent dynamics as a design principle. Pre-training on static corpora treats language as exogenous, producing homogenization effects (Jiang et al., 2025; Yakura et al., 2024, lines 296–303). RLHF optimizes against a frozen reward model vulnerable to the Goodhart dynamics that alignment faking research exposes (Greenblatt et al., 2024; Sheshadri et al., 2025, lines 320–326). Evaluation remains predominantly static leaderboard rankings (Alzahrani et al., 2024, lines 330–335). Where multi-agent considerations appear in foundation model research, they typically take the form of inference-time coordination. This leaves the training distribution static and other agents are accounted as absorbed into the state space.
>
> We hope that our response addresses the concerns raised by the reviewer and we welcome the reviewer to raise any remaining concerns, which we will be happy to discuss further.

---

> > ### Author Rebuttal · Reviewer_VnQ2 · 2026-04-01
> >
> > The rebuttal has addressed my concerns by providing more explanations on the design principle and solipsistic assumptions.

---

### Official Review · Reviewer_cp56 · 2026-03-12

**Significance:** 3
**Argument Clarity:** 3
**Rating:** 4
**Confidence:** 3

**Questions:**

1.  Could you provide a more concrete, localized example of how a standard ML pipeline (e.g., RLHF or PPO for a language model) might implement "Institutions as Design Primitives"? What does a specific algorithmic intervention look like in this context?

2.  Given that the phenomena you describe (train-test-deploy gap, endogenous non-stationarity) are already visible in current systems like recommendation engines and algorithmic pricing, is the focus on "superintelligence" strictly necessary for your position? Would the paper's impact be strengthened by framing it around the immediate trajectory of "agentic AI" rather than a theoretical superintelligence?

**Alternative Views Section:**

Yes

**Compliance With Llm Reviewing Policy A Conservative:**

Affirmed.

**Discussion Potential:**

3

**Paper Summary:**

This position paper argue that the central challenge in AI development is shifting from mere capability (task-solving) to coexistence. The author critique the dominant "solipsistic" AI research paradigm, which assume the environment is exogenous and stationary, and treats other agents merely as background features to be predicted.

The paper posits that deploying highly capable optimization systems (termed "solipsistic superintelligence") in the real world induces "endogenous non-stationarity". Other agents—including humans, institutions, and algorithms—strategically adapt to the AI's presence. This creates a "train-test-deploy gap," where aggressive optimization against historical data accelerates the obsolescence of those very patterns, a phenomenon the authors call the "self-undermining property".

The authors further contend that unilateral prediction cannot solve this issue due to epistemic limits and fundamental societal legitimacy constraints. Ultimately, the paper advocates for a "multi-agent-first" research paradigm, calling for dynamic evaluation testbeds, treating institutions as design primitives, and explicitly preserving human agency.

**Position:**

Yes

**Position In Title:**

Yes

**Related Work:**

3

**Strengths And Weaknesses:**

Strengths:

1.  The position is relevant to the ICML community. As foundation models scale and transition into autonomous agents acting in shared digital and physical spaces, the limitations of single-agent optimization paradigms are becoming a bottleneck.

2.  The formulation of the "solipsistic trap" and the formalization of "endogenous non-stationarity" leading to a "train-test-deploy gap" are compelling. It bridges the gap between classic reinforcement learning assumptions (MDPs) and the reality of Markov games in deployments.

3.  The appendix provides a synthesis of real-world evidence across behavioral (e.g., GPS reliance), institutional (e.g., telematics), and algorithmic (e.g., flash crashes, algorithmic pricing) adaptations, effectively grounding the claims.

Weaknesses:

While the paper successfully diagnoses the problem, the proposed multi-agent-first agenda remains highly abstract. Concepts like "Institutions as Design Primitives" are conceptually sound but lack concrete mathematical or architectural examples of how an ML practitioner should alter their current training loop today.

**Support:**

3

---

> ### Author Rebuttal · Authors · 2026-03-31
>
> We thank the reviewer for their careful engagement with our paper. We are glad to see that the reviewer recognizes the relevance of the position to the ICML community, the value of the solipsistic trap, and the empirical grounding across behavioral, institutional, and algorithmic channels in Appendix B. Below we provide a response to the reviewers' concerns.
>
> We agree that Section 5 would benefit from more concrete guidance on how the multi-agent first agenda translates into practice and we will revise it accordingly. We want to emphasize that we are not proposing a multi-agent training methodology. We are proposing a design principle where strategic interdependence is treated as foundational to how systems are evaluated, deployed, and governed. The methodological anchor is dynamic evaluation, where test distributions are generated by adaptive counterparties rather than fixed benchmarks (Section 5.1, lines 330–344). Institutions and human agency then serve as the two context dependent pillars that shape how cooperative outcomes are achieved in specific deployment settings. That said, this design principle does have concrete instantiations within existing pipelines. Here are the few examples:
>
> - Reinforcement Learning with Evolving Rubrics (RLER) modifies standard RLHF by replacing the frozen reward model with structured rubrics that co-evolve as the agent improves. It focuses on scoring dimensions such as tone calibration, contextual memory, policy adherence, and adaptation to user behavioral shifts. This directly prevents the Goodhart dynamics identified in Section 4.2 (lines 320–326). Recent work further validates this direction: Rubrics as Rewards (2025), DR-Tulu (2025), and OpenRubrics (2025) demonstrate RL beyond verifiable domains using evolving evaluation criteria. Evolving rubrics function as a computational institution because they restructure incentives dynamically, which is precisely what Section 5.2 (lines 345–363) calls for.
> - Training agents within multi-agent simulations such as Concordia (Vezhnevets et al., 2023) or Melting Pot (Leibo et al., 2021), and optimizing for collective outcomes rather than individual task metrics provides another concrete path. Concordia produces auditable artifacts including event logs, dialogue, and state trajectories that enable evaluation of character consistency, global coherence, and psychological plausibility of social modeling.
> - In agentic marketplaces (Tomašev et al., 2025b; Shahidi et al., 2025), agents interact through defined protocols for bidding, reputation, and communication. These protocols function as institutional constraints, and they can themselves be co-learned via mechanism design objectives (Yang et al., 2022).
> - Finally, applying RL to prediction market forecasting provides an inherently non-stationary and reflexive multi-agent signal that resists Goodhart dynamics because ground truth is determined by events rather than the metric itself, bridging performative prediction (Perdomo et al., 2020, Appendix A.2) and practical RL.
>
> On the superintelligence framing, we will revise our paper to make it more clear. The key rationale behind our framing is that cooperation is not a task to be solved by a sufficiently powerful optimizer. It is an equilibrium property emerging from how multiple agents navigate their strategic interdependence (Key Claim 1, lines 103–108, 146–149). We agree that endogenous non-stationarity at deployment time already applies to current agentic systems, but our position is that nothing changes even with superintelligence. The dominant paradigm implicitly assumes that scaling capability will eventually deliver cooperative outcomes just as it delivers better performance on math or coding. We use the superintelligence framing precisely to contest this assumption: no amount of capability resolves the problem because the very act of deploying a powerful optimizer reshapes the strategic landscape it was trained to navigate. The self-undermining property (Proposition A.4) formalizes this and shows that scaling exacerbates the gap. We argue that even multi-agent training will not solve the problem we pose if it is not accompanied by dynamic evaluation. The agentic AI framing is complementary to ours: current agentic systems provide empirical proof that the problem is real today, while the superintelligence framing establishes that it remains structurally irresolvable regardless of capability. We will revise Section 1 and the Scope paragraph (lines 110–125) to make this two level argument explicit.
>
> We welcome the reviewer to engage in further discussion and are happy to address any outstanding concerns.

---

> > ### Author Rebuttal · Reviewer_cp56 · 2026-04-02
> >
> > Thanks for the response and clarification. I maintain my original positive rating of 4.

---

### Official Review · Reviewer_Wad9 · 2026-03-12

**Significance:** 3
**Argument Clarity:** 3
**Rating:** 4
**Confidence:** 4

**Questions:**

How do you see the results of Tonini and Galke (https://arxiv.org/pdf/2508.15510) that show evidence perhaps contrary to the position taken in this paper, that competitive pressure seems to lead to cooperation . They study both one-shot and repeated iterated prisoners dilemma. There are also different results on colluding LLMs (e.g., Fish et al., 2024, Lin et al., 2024) in certain pricing games. Furthermore, from the MARL side Bertrand et al., 2025 show that self-play learners can collude in iterated prisoners dilemma. These papers seem to make the alternative view points stronger.

References:

- Fish, S., Gonczarowski, Y. A., and Shorrer, R. I. Algorithmic collusion by large language models. arXiv preprint arXiv:2404.00806, 7, 2024
- Lin, R. Y., Ojha, S., Cai, K., and Chen, M. Strategic collusion of llm agents: Market division in multi-commodity competitions. In Language Gamification-NeurIPS 2024 Workshop, 2024
- Bertrand, Q., Duque, J. A., Calvano, E., and Gidel, G. Self-play q-learners can provably collude in the iterated prisoner’s dilemma. In International Conference on Machine Learning, 2025

**Alternative Views Section:**

Yes

**Compliance With Llm Reviewing Policy A Conservative:**

Affirmed.

**Discussion Potential:**

3

**Final Justification:**

The rebuttal, especially the last paragraph helped to clarify the authors intended message/position which was not clear in their submitted version and thus I am increasing my score. I think this position paper might be beneficial to the ICML research community.

**Paper Summary:**

The paper argues that solipsistic super-intelligence (i.e., agents that are extremely capable and trained only on "stationary environments", without the presence of other agents are unlikely to be cooperative. Hence, they argue that "multiagent" training must be in the forefront of any agentic training procedure. They also highlight alternative view points on potential standard single agent approaches that can fix the issue (such as scaling, training them to cooperate and that there are no reasons for alarm as current AI systems have not exhibited catastrophic coordination failures).

**Position:**

Yes

**Position In Title:**

Yes

**Related Work:**

2

**Strengths And Weaknesses:**

### Strengths

- They argue their position from various angles, e.g., test-train gaps, exogenous vs endogenous non-stationarity etc, which helps to see the issue from a wider lens, especially from a theoretical side.
- They generally cover important alternative viewpoints, but in my opinion there seems to be alternative viewpoints that seem to be strong against their position (see Questions).
- Overall it is a relevant issue for the ICML community.

### Weaknesses

- It is reasonably well know that multiagent sytems have their own failure modes (e.g., Hammond et al. 2025) and keeping multiagent training on the forefront is an important problem in its own right, irrespective of their position.
- They do not perform any additional experiments to support their position, which in itself is not a problem, perhaps these experiments could have helped support their theoretical arguments more strongly.

**Support:**

2

---

> ### Author Rebuttal · Authors · 2026-03-31
>
> We thank the reviewer for their efforts in reviewing our work and for providing constructive feedback.
>
> We first offer an important clarification on the reviewer’s summary of our work:  This paper does not argue that multi-agent training must be at the forefront of agentic training procedures. Rather, we propose a multi-agent design principle: a way of thinking about building AI that treats strategic interdependence as foundational to evaluation, deployment, governance, and architecture. Multi-agent training can be considered as one possible instantiation, however, our core contribution is the argument that the solipsistic paradigm, which treats the world as a fixed backdrop, cannot deliver the cooperative outcomes deployed by AI. As we emphasize in the paper, cooperation is not a task to be solved by a sufficiently powerful optimizer. It is an equilibrium property that emerges from how multiple agents navigate their interdependence (Key Claim 1, lines 103–108, 146–149). We would go further to argue that even multi-agent training will not solve the problem we pose if it is not accompanied by dynamic evaluation where test distributions are generated by adaptive counterparties rather than fixed benchmarks. Below, we address specific concerns raised by the reviewer.
>
> We explicitly acknowledge multi-agent failure modes in Section 6, Argument 1 (l. 385–420), stating that decentralization alone guarantees nothing (Ostrom, 1990; Hardin, 1968) and also cite Hammond et al. (2025). We never claim that multi-agent approaches are inherently safe. Our claim is that AI systems built under solipsistic paradigm encounter multi-agent dynamics at deployment whether or not they were designed for them (lines 399–405). Regarding multi-agent research being important irrespective of our position, this is in fact our position. Our contribution is the formal diagnosis of why the dominant paradigm structurally cannot deliver cooperative outcomes. We identify endogenous non-stationarity (Definition 3.1), the self-undermining property (Proposition A.4), and equilibrium selection risk (Proposition A.9) as the root causes. As we note at lines 88–91, cooperative AI insights remain peripheral to the central scaling pathway of training solitary AI models. We argue this peripherality is untenable.
>
> Regarding lack of experiments, this is a position paper and our arguments are grounded in extensive empirical evidence across all three channels of structured adaptation in Section 3.3 and Appendix B (lines 880–1044), including GPS induced cognitive degradation (Dahmani and Bohbo, 2020), writing homogenization from auto-complete (Arnold et al., 2020), insurance telematics feedback loops (Husnjak et al., 2015), emergent algorithmic collusion (Calvano et al., 2020), advertising auction dynamics (Banchio and Skrzypacz, 2022), and cryptocurrency bot interactions (Daian et al., 2020). If the reviewer has particular experiment in mind that would strengthen our  argument, we welcome specific suggestions and we will discuss and carefully consider them.
>
> We thank the reviewer for sharing pertinent references which we believe are largely consistent with our position. The collusion results from Fish et al. (2024), Lin et al. (2024), and Bertrand et al. (2025) show solipsistic agents converging on competitive outcomes that benefit colluding parties at the expense of third parties. We use the term cooperation in our paper broadly to mean the problem of selecting good equilibrium. Collusion that harms third parties is a degraded equilibrium, exactly the equilibrium selection risk formalized in Proposition A.9. The AI system in Tonini and Galke operates under conditions that our paper explicitly identifies as insufficient for real deployment. Cooperation in iterated prisoner's dilemma is consistent with classical results (Axelrod, 1984; Nowak, 2006) because the game satisfies all conditions that sustain cooperation by construction: repeated interaction with identifiable partners, reputation mechanisms, and stationary rules. We note in Section 6 (lines 421–438) that deployment environments violate these conditions because interactions are anonymous, systems can be retrained or rebranded, and externalities fall on parties outside the game. Further, all cited papers study simplified games with no institutional, behavioral, and legitimacy dimensions (lines 118–125, 275–295). Further, the reviewer's references demonstrate an important point: these results are context dependent, with the outcome determined by game structure, training procedure, and initial conditions. This is exactly our point as we argue for a design paradigm that treats context, strategic interdependence, and institutional structure as first order concerns. We will incorporate discussion of these references in a revised Section 6.
>
> We hope that our responses provide helpful clarifications and we welcome further discussion if there still remains any outstanding concerns.

---

> > ### Author Rebuttal · Reviewer_Wad9 · 2026-04-03
> >
> > I see what the authors were going for and my concerns have been addressed. I hope that the authors would add these points in their final draft, especially more in line with the discussion wrt Collusion and cooperation etc.

---

### Decision · Program_Chairs · 2026-04-30

**Decision:**

Accept (regular)

**Comment:**

This paper argues that we must consider AI in the context of a non-stationary world where benchmarks and training data cannot keep up.  The paper argues that we should move to a multi-agent research paradigm that encourages strategic interdependence.


The reviewers liked many of the ideas in the paper but wished they were more concrete. Some of the examples that came up in review will help to clarify the argument. There were some calls for empirical evidence, but I think this paper is strong without such evidence (nor is it clear what experiments exactly would strengthen the paper).  There were questions about the exact situations where these ideas apply (single agent? Sub-superintelligence?), which the authors handled well.


AC note: I noticed that the authors may be using multiple terms for the same thing — it may help with clarity to keep wording consistent.  e.g. is “irreducible interdependence” the same as “strategic interdependence”? Your paper’s impact hinges on its clarity.